# Horizontal Combination of MEK and PI3K/mTOR Inhibition in BRAF Mutant Tumor Cells with or without Concomitant PI3K Pathway Mutations

**DOI:** 10.3390/ijms21207649

**Published:** 2020-10-16

**Authors:** Dominika Rittler, Eszter Molnár, Marcell Baranyi, Tamás Garay, Luca Hegedűs, Clemens Aigner, József Tóvári, József Tímár, Balázs Hegedűs

**Affiliations:** 12nd Department of Pathology, Semmelweis University, H-1091 Budapest, Hungary; rittlerdomi@gmail.com (D.R.); m.molnareszter@gmail.com (E.M.); baranyi2marci@gmail.com (M.B.); jtimar@gmail.com (J.T.); 2Faculty of Information Technology and Bionics, Pázmány Péter Catholic University, H-1083 Budapest, Hungary; garayt@gmail.com; 31st Department of Internal Medicine and Oncology, Semmelweis University, H-1091 Budapest, Hungary; 4Department of Thoracic Surgery, Ruhrlandklinik, University Duisburg-Essen, D-45239 Essen, Germany; Luca.hegedues@rlk.uk-essen.de (L.H.); clemens.aigner@ruhrlandklinik.uk-essen.de (C.A.); 5Department of Experimental Pharmacology, National Institute of Oncology, H-1122 Budapest, Hungary; jtovari@yahoo.com

**Keywords:** combination therapy, BRAF, PTEN, PI3K, BEZ235, selumetinib

## Abstract

The RAS/RAF and PI3K/Akt pathways play a key regulatory role in cancer and are often hit by oncogenic mutations. Despite molecular targeting, the long-term success of monotherapy is often hampered by de novo or acquired resistance. In the case of concurrent mutations in both pathways, horizontal combination could be a reasonable approach. In our study, we investigated the MEK inhibitor selumetinib and PI3K/mTOR dual inhibitor BEZ235 alone and in combination in BRAF-only mutant and BRAF + PI3K/PTEN double mutant cancer cells using short- and long-term 2D viability assays, spheroid assays, and immunoblots. In the 2D assays, selumetinib was more effective on BRAF-only mutant lines when compared to BRAF + PI3K/PTEN double mutants. Furthermore, combination therapy had an additive effect in most of the lines while synergism was observed in two of the double mutants. Importantly, in the SW1417 BRAF + PI3K double mutant cells, synergism was also confirmed in the spheroid and in the in vivo model. Mechanistically, p-Akt level decreased only in the SW1417 cell line after combination treatment. In conclusion, the presence of concurrent mutations alone did not predict a stronger response to combination treatment. Therefore, additional investigations are warranted to identify predictive factors that can select patients who can benefit from the horizontal combinational inhibition of these two pathways.

## 1. Introduction

The RAS/RAF/MEK/ERK and PI3K/Akt/mTOR pathways have a crucial regulatory role in mammalian cells by controlling the cell growth, proliferation, differentiation, protein synthesis, motility, metabolism, and survival [1,2]. Therefore, oncogenic alterations of key proteins in this pathways cause constitutive activation independently from upstream signals and contribute to the development, maintenance, and progression of cancers [3,4,5].

One of the most frequently mutated gene is BRAF, a member of a family with three specific isoforms (ARAF, BRAF, and CRAF). It has oncogenic mutations in 50–60% of melanomas, 5–15% of colon cancers, and 3–5% of lung adenocarcinomas. The V600E alteration is the most prevalent present in 80–90% of BRAF mutant melanomas, in 50% of BRAF mutant lung adenocarcinomas, and in 90% of BRAF mutant colorectal cancers [6,7,8]. However, non-V600 BRAF mutations are also relevant in several cancer types and lead to the oncogenic activation of the RAS/RAF pathway [9,10]. Further downstream mutations in RAF/MEK/ERK cascade (MEK1/2, Erk1/2) are rarely detected in melanoma (4–8%), colorectal cancer (3–4%), or lung adenocarcinoma (0.5–1.5%) [11]. Of note, MEK1/2 are the exclusive kinase of Erk1/2, therefore the inhibition of MEK1/2 could be an efficient approach to inhibit the constitutively activated pathway [12].

Mutations of the PTEN and PI3K genes are the most frequent oncogenic alterations in the PI3K/Akt cascade. PTEN is a key antagonist of PI3K. PTEN mutation is detected in 17% of melanoma, 10% of colorectal cancer, and 4% of lung adenocarcinoma cases [11]. The loss of function mutation of PTEN leads to the enhanced activity of Akt and the PI3K/Akt pathway [13,14]. Mutations of class IA PI3K members, especially the catalytic subunit p110α (PI3KCA) are the most frequently described in cancers [11,15,16]. PI3KCA and PI3KR1 mutations are detected in less than 5% of melanoma and lung adenocarcinoma and 10–20% of colorectal cancer cases [11]. Both the loss of PTEN and PI3K activating mutations contribute to the accumulation of PIP3, a second messenger for the activation of Akt by PDK1. Of note, the complete activation of Akt requires phosphorylation by mTORC2 as well [17].

Mutations in both signaling pathways were described in different tumor types. KRAS or BRAF alterations with concomitant PI3KCA mutation were found in colorectal carcinomas in 7–9% and 1–12%, respectively [11,18,19,20]. Additionally, in lung adenocarcinoma and melanoma, 3–6% and 0.14–3% of the BRAF mutant cases harbor concomitant PI3KCA mutation, respectively [11,19]. Furthermore, other PI3K/Akt pathway mutations (such as PTEN, mTOR, PIK3R1, and NFKB1) can co-occur in melanoma with BRAF (17–20%) or NRAS (9%) mutations [21,22,23].

The RAS/RAF/MEK/ERK and PI3K/AKT/mTOR pathways have been previously described to be able to cross-regulate each other at several points, resulting either activation or inhibition [24]. Some of the most important interactions are the PI3K direct activation by RAS [25], TSC2 inactivation by ERK [26] and RAF regulation by Akt [27]. The inhibition of MEK was described to initiate the activation of the PI3K/Akt pathway by enhancing the Akt activation [9,28,29], in addition, PTEN null mutation was shown to contribute to the BRAF inhibitor resistance [30]. Furthermore, BRAF or MEK inhibition resistance can develop via upregulation of upstream elements including RAS or RTKs [31,32]. Based on all these mechanisms, the dual targeting of both pathways seems to be necessary for the successful treatment of certain cases.

The combination of systemic therapies including chemotherapy, targeted therapy, or immunotherapy agents is able to interfere with the development of therapy resistance [33]. For example, BRCA mutant breast cancers show an improved response against a combination of PARP inhibitors and chemotherapy [33,34]. Furthermore, synergism between the combined agents may facilitate the use of smaller doses and thus may result in fewer side effects.

Selumetinib is a potent, highly selective MEK1/2 inhibitor [35,36] with an inhibitory effect in various cancer types, especially in BRAF and/or RAS mutant tumors [36,37,38,39]. It was also studied in numerous clinical trials alone or in combination with other compounds [40,41,42]. Recently, selumetinib was clinically approved by the FDA for the treatment of pediatric neurofibromatosis type 1 plexiform neurofibromas (NCT01362803, [43]). In our study, we used a MEK inhibitor due to the fact that one of our BRAF mutant cell line has non-V600 BRAF mutation, therefore, there is no specific BRAF inhibitor.

Currently, there are four isoform specific PI3K inhibitors approved by the FDA for different tumors, however, they show low efficacy against PTEN mutant cancers [14]. There are more than ten PI3K/mTOR dual inhibitors in Phase I-II clinical trials, however, due to low efficacy and high toxicity, none of them have reached Phase III trials. We selected one of the most studied dual inhibitors, BEZ235, as three of our cell lines carry PTEN mutation. BEZ235 is a dual ATP-binding PI3K/mTOR inhibitor targeting both mTORC1 and mTORC2 [44,45]. Numerous preclinical studies described a successful inhibitory potential against a variety of cancer types (melanoma, colorectal, and lung adenocarcinomas) [46,47,48]. However, the cancer inhibitory effect of BEZ235 was limited and tolerability was low in the majority of clinical trials and thus currently there are no clinical indications for this drug [49,50]. While currently there is no further oncological development of BEZ235, there is an ongoing Phase III clinical trial with COVID-19 patients investigating the potential of BEZ235 (clinicaltrials.gov: NCT04409327).

Dual blockade of both RAS/RAF and PI3K/Akt pathways may be able to decrease the side effects and minimize the risk of development of resistance as it was previously described that combination of MEK—and a PI3K/Akt pathway inhibitor was able to increase progression free survival [51,52,53]. Accordingly, in this study we investigated the inhibitory effect of BEZ235 and selumetinib alone and in combination on BRAF mutant melanoma, colorectal and lung cancer cell lines with or without PI3K/Akt pathway mutations.

## 2. Results

### 2.1. Sensitivity of the Cancer Cell Lines in Short-Term Treatments

Short-term SRB assay was applied in order to investigate the growth inhibitory effects of the inhibitors on the cell lines in different concentrations. The MEK inhibitor selumetinib had a higher inhibitory effect on the BRAF mutant, PI3K/PTEN wild type (blue marked) cell lines when compared to the BRAF + PI3K/PTEN mutant (green) cells (Figure 1a,c, Appendix A). Interestingly, two cell lines of the double mutant group (HT29, SW1417) with PI3K mutation were the most resistant to selumetinib treatment. In addition, the PI3K/mTOR dual inhibitor BEZ235 seemed not to have mutation specific inhibitory effect on the cells (Figure 1b,d, Appendix A).

### 2.2. Long Term Effect of the Inhibitors Alone and in Combinations

Colony formation inhibition of the drugs was analyzed alone by 10 day-long clonogenic SRB assay. Selumetinib seemed to be more effective on BRAF-only mutant (blue) cells, similar to the short term viability assay results (Figure 2a,c). Additionally, SW1417 cell line was still the most resistant to selumetinib treatment (Figure 2a). Of note, BRAF + PI3K/PTEN mutant (green) cells are more resistant in average to selumetinib than only BRAF mutant cells, even without colon cell line results (Appendix A). BRAF + PI3K/PTEN mutant cells were more sensitive to BEZ235 when compared to BRAF mutant cell lines (Figure 2b,d). The combinatory index (CI) was calculated from the average of the 10 day-long treatment data. In the majority of cell lines, the drugs had only additive effects (CI ≈ 1) (Figure 3a). Notably, in two cell lines (SW1417 and WM239) several combination concentrations resulted in synergism (CI < 1). Furthermore, the average of the CI values across all concentrations confirmed that the CI values were significantly under 1 in case of WM239 and SW1417 cells, (Figure 3b).

### 2.3. Effect of the Inhibitors on Spheroid Growth

In order to validate our 2D findings in spheroid models, the cell lines with synergism in the combination treatments (WM239, SW1417; CI < 1) and two cell lines with only additive effects (A375, HT29, CI ≈ 1) were selected for three-dimensional experiments. Representative pictures demonstrate the effect of the treatments (Figure 4a and Appendix A). The volume growth of the spheroids was significantly inhibited upon treatment with selumetinib and selumetinib + BEZ235 in all four cell lines (Figure 4b). The combination seemed to be slightly more effective than the single selumetinib treatment in A375 and HT29 spheroids. Importantly, in SW1417, the combination treatment was significantly more effective when compared to the single treatments. Furthermore, CCK8 viability measurement confirmed the spheroid volume results (Figure 4c).

### 2.4. Apoptosis Induction and Decreased Signaling Upon Treatment

Western blot analysis was performed for 48 h to detect apoptosis induction and proliferation change upon treatment with selumetinib, BEZ235, or combination (Figure 5a and Appendix A). Based on the increased cleaved PARP level, selumetinib and combination therapy were able to induce apoptosis in case of A375 and WM239. Interestingly, PARP cleavage was not present upon treatment in the SW1417 and HT29 lines. Furthermore, PCNA level decreased upon treatment with selumetinib or with the combination in A375 and WM239 cell lines but not in SW1417 and HT29 cells. Additionally, we analyzed the cell cycle distribution by image cytometry after treatments. We found an increased proportion of cells in the subG1 phase indicating apoptosis induction after selumetinib and combination treatments in the A375 cell line (Appendix A). Furthermore, in WM239, the ratio of cells in S and G2/M phase clearly decreased by the same treatments. Importantly, while we could not demonstrate apoptosis induction by PARP cleavage or cell cycle distribution changes in the HT29 and SW1417 cells, the total cell number was strongly reduced and particularly after the combination treatment (Appendix A). To investigate the activation of the two signaling cascade, further immunoblot assays were performed for the phosphorylation of Akt, S6, and Erk proteins (Figure 5b and Appendix A). A decrease in *p*-S6 level was measured after BEZ235 and combination treatments; however, a significant change was detected only in the combination. Erk activation decreased especially upon treatment with selumetinib or combination, however, it was significant only in the SW1417 cells. Interestingly, Akt activation did not changed upon any treatments, except for the SW1417 cell line, where *p*-Akt level decreased after treatment with BEZ235 and was significantly reduced with the combination treatment.

### 2.5. Baseline Activation of Certain Proteins in the Five Double Mutant Cell Lines

To investigate whether the sensitivity of the BRAF + PI3K/PTEN mutant cell lines (SKMEL28, A2058, WM239, HT29, SW1417) correlate with baseline activation of signal pathway proteins we determined differences in the activation of S6, Akt, Erk, as well as expression of MET and EGFR proteins in all five double mutant cells via western blot. EGFR and MET expression were detected in colon cell lines (HT29, SW1417) with BRAF + PI3K mutations (Appendix A). Interestingly, phospho-Erk level was much higher in PTEN mutant and selumetinib sensitive cell lines than in others (Appendix A). Additionally, enhanced p-Akt was detected in PTEN mutant cells, in contrast, *p*-S6 level was similar in all cell lines, except for in HT29.

### 2.6. The Efficacy of the Combination Therapy Was Recapitulated in the SW1417 Xenograft Model

SW1417 cells were injected subcutaneously into NOD-SCID female mice to validate our in vitro results in vivo (Figure 6). The single treatment with selumetinib (25 mg/kg) and BEZ235 (15 mg/kg) had a significant inhibitory effect on tumor growth compared to the control group. However, combination was even more effective than the single treatments as measured by the volume of the tumor (Figure 6a). Tumors were dissected and weighted at the end of the experiment and the combination resulted in the strongest tumor weight reduction (Figure 6c,d). While there was no significant difference between the treatment groups using one-way ANOVA test, by comparing treatment groups to control via unpaired t-test showed that the tumor mass decrease was significant only after the combination treatment.

## 3. Discussion

Targeted therapy for BRAF (especially BRAF V600E) mutant tumors with BRAF and/or MEK inhibitors is currently the standard-of-care treatment, however, resistance to therapy inevitably appears [31,32]. The cross-talk between RAS/RAF and PI3K/Akt pathways and the concomitant PI3K pathway mutations in BRAF mutant tumors suggest that horizontal combination inhibition of these cascades might increase treatment efficacy [24]. MEK1/2 serves as a gatekeeper in the RAS/RAF pathway being an exclusive activator of Erk, therefore, MEK inhibition is a promising way to successfully block the downstream elements of the cascade [12]. Additionally, the inhibition of both mTORC1 and mTORC2 is critical for effective blocking of PI3K/Akt pathway which cannot be achieved by rapamycin but with BEZ235 [44,54]. In this study, we presented the effectiveness of the MEK inhibitor selumetinib and the PI3K/mTOR inhibitor BEZ235 alone and in combination on BRAF and BRAF + PI3K/PTEN mutant human cancer cell lines from different types of cancer (melanoma, lung and colorectal adenocarcinoma), in line with the current basket-type clinical trial design for BRAF mutant tumors [55]. 

The short- and long-term effectivity of selumetinib and/or BEZ235 was tested previously on different types of tumor cell lines with similar results (Figure 1 and Figure 2) [29,46,56,57,58]. Sweetlove and co-workers presented in a short-term experiment that the selumetinib and BEZ235 combination has nearly additive effect on most BRAF mutant melanoma cell lines, similar to our long-term assay results [57]. In our study, we found the combinations with synergistic inhibitory effect on two cell lines (one melanoma and one colon) harboring BRAF and concomitant PI3K pathway mutations but not all the cell lines in this mutational group (Figure 3). Interestingly, the combination of these inhibitors was also found to be synergistic in certain BRAF wild type and V600E mutant, EGFR, and KRAS + PI3KCA mutant cell lines both in short- and long-term examinations [46,58].

Regarding apoptosis induction in two of four cell lines studied, selumetinib was able to induce apoptosis but the combination with BEZ235 showed a stronger effect (Figure 5 and Appendix A). Previous studies also described enhanced apoptosis induction with the combination [59,60,61]. Interestingly, we could not detect cleaved PARP in SW1417 and HT29 cell lines despite their high sensitivity to the treatments. Additionally, PCNA, a cell proliferation associated protein was decreased upon treatment with selumetinib and combination in A375 and WM239 cells but not in SW1417 and HT29 cells (Figure 5 and Appendix A). Furthermore, based on cell cycle analysis, the increased ratio of cells in subG1 phase was observed in the A375 cell line and decreased G2/M and S phase proportions in WM239 upon treatment with selumetinib and combination (Appendix A) in line with previous studies [29,59]. The lack of PARP cleavage and cell cycle distribution alterations during the growth inhibition of the two colon cancer cell line might be due to the TP53 mutation in these cell lines [62,63].

In order to compare the impact of PTEN or PI3K mutations the baseline activation of Akt, S6 and Erk proteins were measured in all five BRAF + PI3K/PTEN mutant cell lines (SKMEL28, A2058, WM239, HT29, SW1417). In line with previous observations, the p-Akt level was higher in BRAF + PTEN mutant cell lines when compared to BRAF + PI3K mutant cells (Appendix A) [64]. Regarding the impact of treatment on PI3K/Akt and RAS/RAF pathways upon treatment we observed strong inhibitory effect of selumetinib on Erk activation alone and in combination with BEZ235, in line with previous studies [56,57,61]. Furthermore, we found that baseline Erk phosphorylation is higher in cells with higher selumetinib sensitivity when compared to the resistant lines (Appendix A). As expected, *p*-S6 level decreased upon treatment with BEZ235 alone in all four cell lines (A375, WM239, HT29, SW1417) but there was a significantly stronger effect with BEZ235 + selumetinib, similarly to previous observations [59]. Selumetinib was found to have an inhibitory effect on S6 activation after 24h treatment in an earlier study on BRAF mutant melanoma cell lines [57], which may contribute to the enhanced effect of the combination when compared to single treatment. Previous studies showed that BEZ235 is able to inhibit Akt activation as well, especially in combination with selumetinib [61,65]. Interestingly, we found decreased p-Akt level only in the most sensitive cell line, SW1417, upon treatment with the BEZ235 and selumetinib combination. Furthermore, high EGFR and MET expression was detected in the BRAF + PI3K mutant colon cell lines (Appendix A). It is known that EGFR expression is more pronounced in colon cells compare to melanoma that may explain the lower sensitivity of the investigated colon cell lines to MEK inhibition [66]. Interestingly, in case of MET expression, only the tissue origin of the cells does not explain the differences among melanoma and colon cell lines (Appendix A) [67]. Of note, our study was not able identify the precise predictive role of the individual mutations. Additional studies using isogenic cell lines pairs with and without the specific mutations are warranted to determine the impact of individual mutations in combination treatments.

To validate our two-dimensional results, three-dimensional spheroids and subcutaneous xenografts were also treated. The two cell lines with synergistic effect in the combination treatment (WM239, SW1417) and two lines with additive effects (A375, HT29) were used in spheroid growth and viability assays (Figure 4 and Appendix A). Cell lines showed different sensitivity against the treatments, but the combinational treatment was significantly more effective than single treatment only in SW1417 cells, in line with the two-dimensional experiments. Interestingly, in a previous study, the combination treatment with selumetinib and a higher BEZ235 treatment concentration were described to be significantly more effective on HT29 cell line than single treatments [68]. In addition, vemurafenib (BRAF inhibitor) resistant melanoma lines (M229, M238, Pt48) demonstrated higher sensitivity towards BEZ235 + selumetinib therapy previously [61]. Based on our in vitro experiments, we chose the SW1417 cell line, with the highest synergism in combinational treatment, for the xenograft experiment. The combinational treatment was able to decrease tumor growth more successfully than either of the single agents. Previous studies also demonstrated that selumetinib and BEZ235 in combination are more effective than these drugs alone on certain BRAF mutant melanoma cell lines (NZM20) [57], in certain lung cancer cell lines (NCI-H1993, NCI-H1975, NCI-H460) [58], or even in patient-derived xenograft models of colorectal carcinomas [69].

## 4. Materials and Methods

### 4.1. Cell Lines and Reagents

Eight human cancer cell lines were investigated, including five melanoma (A375, WM35, SKMEL28, A2058, and WM239), one lung adenocarcinoma (CRL5885), and two colon adenocarcinoma cell lines (HT29, SW1417) (Table 1). The A375, SKMEL28, CRL5885, A2058, HT29, and SW1417 cells were obtained from ATCC, WM35 and WM239 cells were from the Wistar Institute. Cell lines were cultured in DMEM (Lonza, Switzerland, with 4500 mg/dm^3^ glucose, L-glutamine and pyruvate) supplemented with 10% fetal bovine serum (FBS) (Gibco-BRL Life Technologies, U.K.) and 1% penicillin-streptomycin-amphotericin (Lonza, Basel, Switzerland) at 37 °C and 5% CO_2_ in humidified atmosphere in tissue culture flasks. Selumetinib and BEZ235 compounds were purchased from Selleck Chemicals (Houston, TX, USA). The animal-model experiment was carried out according to the standards and guidelines approved by the Animal Care and Use Committee of the National Institute of Oncology, Budapest, Hungary (PEI/001/2574–6/2015, 12/10/2015).

### 4.2. Sulforhodamine B (SRB) Assay

The short term (72h) antiproliferative effect of the drugs was assessed by SRB assay. Briefly, 5000 cells were seeded in the inner 60 wells of a 96 well plate and incubated for 24 h. Next day, cells were treated with selumetinib and/or with BEZ235 in different concentrations. After a 72-h incubation, cells were fixed to the bottom of the wells with 10% TCA (trichloroacetic acid) and SRB staining was applied. Then, 10 mM Tris-HCl buffer (pH = 7.4) was used to dissolve the bound dye and then the optical density (OD) was detected by a micro plate reader (EL800, Bio-Tec Instruments, Winooski, USA) at 570 nm.

### 4.3. Colony Formation Inhibition and Combinatory Index Evaluation

Clonogenic assay was performed to determine the long term (10 days) inhibitory effect of selumetinib and/or BEZ235. Briefly, cells were seeded at appropriate concentration (250–4000 cell/well) in 24 well plates and incubated until attachment. 24 h later, selumetinib and BEZ235 were added at various concentrations (0.025 µM, 0.05 µM, 0.1 µM, 0.25 µM, 0.5 µM, and 1 nM, 2.5 nM, 5 nM, 10 nM, respectively) alone and in combination. Ten days later, cell monolayers were fixed with 10% TCA and SRB staining and OD measurement was used as described above. Interaction between selumetinib and BEZ235 was tested and CI (combinatory index) values, indicating synergistic, additive and antagonistic effect if CI < 1, CI ≈ 1 and CI > 1, respectively, were calculated with CompuSyn software (ComboSyn Inc., Paramus, NJ, USA) according to Chou and Talalay [71]. 

### 4.4. Spheroid Growth Inhibition

3D investigations were performed to assess the single drug and combinatory effect of the inhibitors on four cell lines (A375, WM239, HT29, and SW1417). U-bottom 96 well plates (Sarstedt, Nümbrecht, Germany) were pre-coated with polyHEMA (5 mg/mL (Sigma-Aldrich, St. Louis, MO, USA) to avoid the attachment of the cells. For spheroid generation, 1000 cells were seeded in the inner 60 wells of the U-bottom 96 well plates and incubated for 96 h In order to achieve spheroid generation of WM239 cell, additional 1% Matrigel (Growth Factor Reduced, Corning, NY, USA) was used, and for HT29, 1% Matrigel and 10 µg/ml collagen (Corning) was applied. Spheroids were treated with the inhibitors alone and in a combination and pictures were taken every third day using 4x objective. The area and radius of the spheroids were measured by ImageJ program and the equivalent sphere volume was calculated (4/3 × π × radius^3^). Additionally, cell viability inhibition of the treatments was detected at the sixth day applying CCK8 (Dojindo, Kumamoto, Japan) for 4 h and then OD was measured at 450 nm.

### 4.5. Immunoblot Analysis

Western blot assay was performed to follow the protein expression and phosphorylation changes upon treatment with selumetinib and BEZ235 alone or in combination in four cell lines (A375, WM239, HT29, and SW1417). The cells were seeded in 6 well plates in a 1−3 × 10^5^ concentration. Upon attachment, the cells were treated and processed for the apoptosis induction (c-/PARP) and proliferation change (PCNA) detection, as well as for the detection of Akt, S6 and Erk protein phosphorylation. The treatment concentrations were determined based on short term viability assay results. Higher treatment concentrations (selumetinib (100 nM), BEZ235 (10 nM)) and a longer period of treatment time (48 h) was applied to determine apoptosis induction and proliferation change in the cells. For the detection of changes in signaling pathway protein activation, lower concentrations (selumetinib (50 nM), BEZ235 (5 nM)) and shorter treatment (4 h) was used. At the end of the treatment, the cellular proteins were precipitated by ice-cold 6% TCA at 4 °C for minimum 1 h. The supernatant medium with detached cells was also collected and precipitated for the 48 h-long treated samples. Of note, in the case of sensitive cell lines, more cells were treated in more parallel wells to provide the sufficient amount of sample for the investigation. Besides, the cell cultures were checked under the microscope before harvesting to ensure they were suitable and sufficient for the experiment. Precipitates were scratched from the bottom, centrifuged at 3500 rpm at 4 °C for 15 min, and TCA was removed from the pellet. A suitable volume of SSP buffer (10% glycerol, 2% SDS, 62.5 mM Tris-HCl pH 6.8, 125 mg/mL urea, 5 mM EDTA, 0.14mg/mL of bromophenol blue and 10 mM dithiothreitol) was used as dissolvent for the pellets. Qubit protein assay kit (Thermo Scientific, Waltham, MA, USA) was applied to measure total protein concentration. For western blot, 20–25 µg protein was loaded on 10% polyacrylamide gels, separated by gel electrophoresis then transferred to PVDF membranes (Thermo Scientific). These primary antibodies were applied in a dilution of 1:1000 at 4 °C, overnight: PARP, PCNA, p-Akt/Akt, *p*-Erk1/2,/Erk1/2 and *p*-S6/S6, PTEN, MET, EGFR (Cell Signaling; #9542, #13110, #4058, #9272, #9101, #9102, #2217, respectively) and anti β-tubulin and GAPDH (Abcam, ab6046, Cell Signaling #5174, respectively) were used as the loading control. Then, secondary HRP-labelled anti-rabbit antibody (Jackson ImmunoResearch, West Grove, PA, USA) was applied in a dilution of 1:10000 for 1 h at room temperature. For visualization, Pierce ECL Western Blotting Substrate (Thermo Scientific) and CL-XPosure Film (Thermo Scientific) were used. Blots were analyzed quantitatively by ImageJ software (Fiji package) after normalization to loading control proteins or Ponceau staining.

### 4.6. Cell Cycle and Cell Number Analysis

Cell cycle analysis was performed based on the DNA content of the cells. Briefly, the cells were treated with selumetinib (100 nM), BEZ235 (10 nM), or both for 72 h in 6-well plates. Then, both supernatant and attached cells were collected and mixed with lysis buffer + DAPI for 5 min at 37 °C. After the application of stabilization buffer, the samples were analyzed by NucleoCounter NC-3000 system (Chemometec, Allerod, Denmark) following the manufacturer’s protocol. In simultaneous parallel experiments, the cell number was determined at the end of the treatment by the NucleoCounter NC-3000 system (Chemometec). After trypsinization, the cells were stained with Acridine Orange and DAPI (Solution 13, Chemometec, 910–3013). Then, 10 µL of each sample was loaded in an 8-well NC-slide and the cells were counted. 

### 4.7. In Vivo Xenograft Experiment

The in vivo experiment was performed at the National Institute of Oncology, Budapest, Hungary, according to the Guidelines for Animal Experiments and was approved for the Department of Experimental Pharmacology (Permission Number: PEI/001/2574–6/2015, 12/10/2015). The subcutaneous xenograft model was established by injecting SW1417 cells (2 × 10^6^) into the flank of NOD-SCID female mice. When the tumors were measurable (13 days after injection), the mice were divided into four groups (selumetinib 25 mg/kg, BEZ235 15 mg/kg, combination, and a control group with vehicle constituents). Treatment solutions were prepared based on the manufacturer instructions (selumetinib: DMSO, PEG300, H_2_O– 5:72:23; BEZ235: NMP, PEG300–10:90) and were applied by oral gavage once daily on weekdays for 17 days. The body weight of the mice was monitored twice weekly. After measurement of the tumors’ diameter by a caliper, tumor volumes were calculated with the formula for the volume of a prolate ellipsoid (width^2^ × length × π/6). At the end of the experiment, the mice were sacrificed, and tumor tissues were removed and weighed.

### 4.8. Statistics

One-way ANOVA and non-parametric Kruskal–Wallis test were used, followed by Tukey’s and Dunn’s post hoc tests to determine statistical differences between treatment groups in immunoblots and in 3D experiments. One-sample t-test was applied to determine the statistical significance of additive and synergistic effect in combination treatments. One-way repeated measures ANOVA followed by Tukey’s post hoc test was applied for the in vivo experiments to assess differences between the groups. Statistical significance was indicated as * *p* < 0.05, ** *p* < 0.01 and *** *p* < 0.001. All statistical analyses were performed in GraphPad Prism 5 (GraphPad Software Inc, San Diego, CA, USA).

## 5. Conclusions

In conclusion, we demonstrated that the concomitant mutations in RAS/RAF and PI3K/Akt pathway does not unequivocally predict synergistic effect of the horizontal combination of inhibitors of MEK and PI3K/mTOR. Further studies are warranted to identify the predictive factors for selecting the tumors that will respond synergistically to combination therapies.

## Figures and Tables

**Figure 1 ijms-21-07649-f001:**
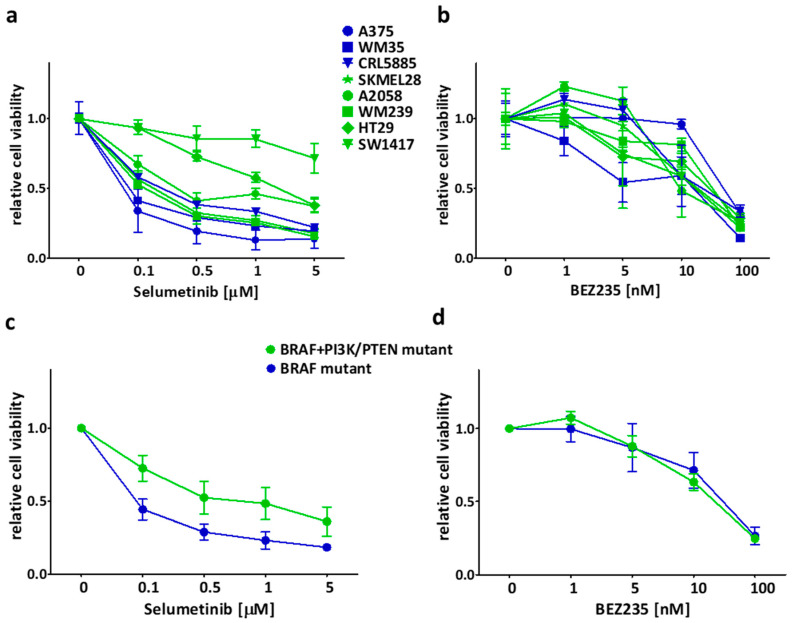
Short term (72 h) effect of selumetinib and BEZ235 on the cell viability (SRB assay). (**a**,**b**) Selumetinib was more effective on BRAF mutant cell lines (blue) than on BRAF + PI3K/PTEN mutant cell lines (green). After treatment with BEZ235, sensitivity difference was not detected in the mutational groups. (**c**,**d**) Average cell viability upon treatment with selumetinib or BEZ235 in the two mutational groups. Data is shown as mean ± SEM from at least three independent experiments.

**Figure 2 ijms-21-07649-f002:**
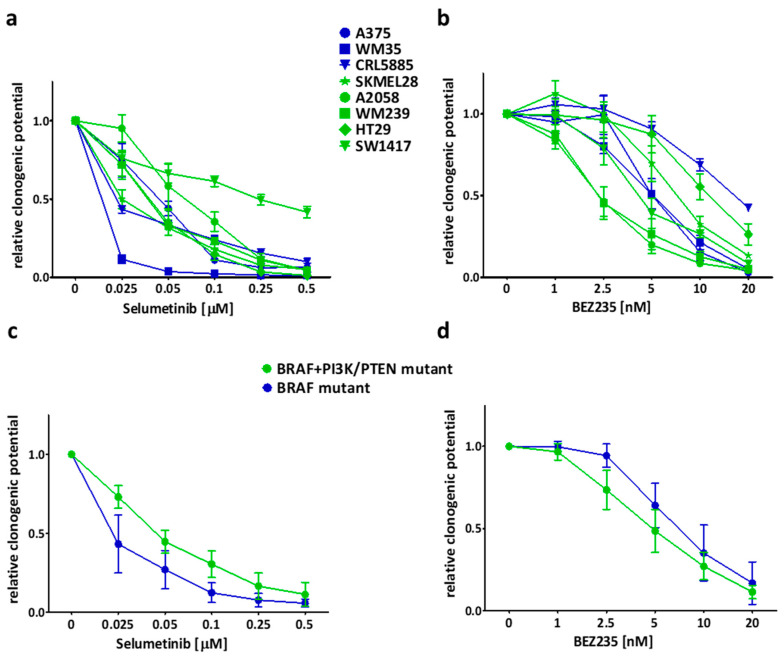
Colony formation inhibitory effect of selumetinib and BEZ235 for 10 days. (**a**,**b**) BRAF mutant (blue marked) cell lines were more sensitive to selumetinib than the BRAF + PI3K/PTEN mutant (green) cells; on the other hand, BEZ235 inhibited the cell lines with BRAF + PI3K/PTEN mutation more effectively than the BRAF mutant cell lines. (**c**,**d**) Average clonogenic potential in the mutational groups after treatment with selumetinib or BEZ235. Data is shown the mean ± SEM from at least three independent experiments.

**Figure 3 ijms-21-07649-f003:**
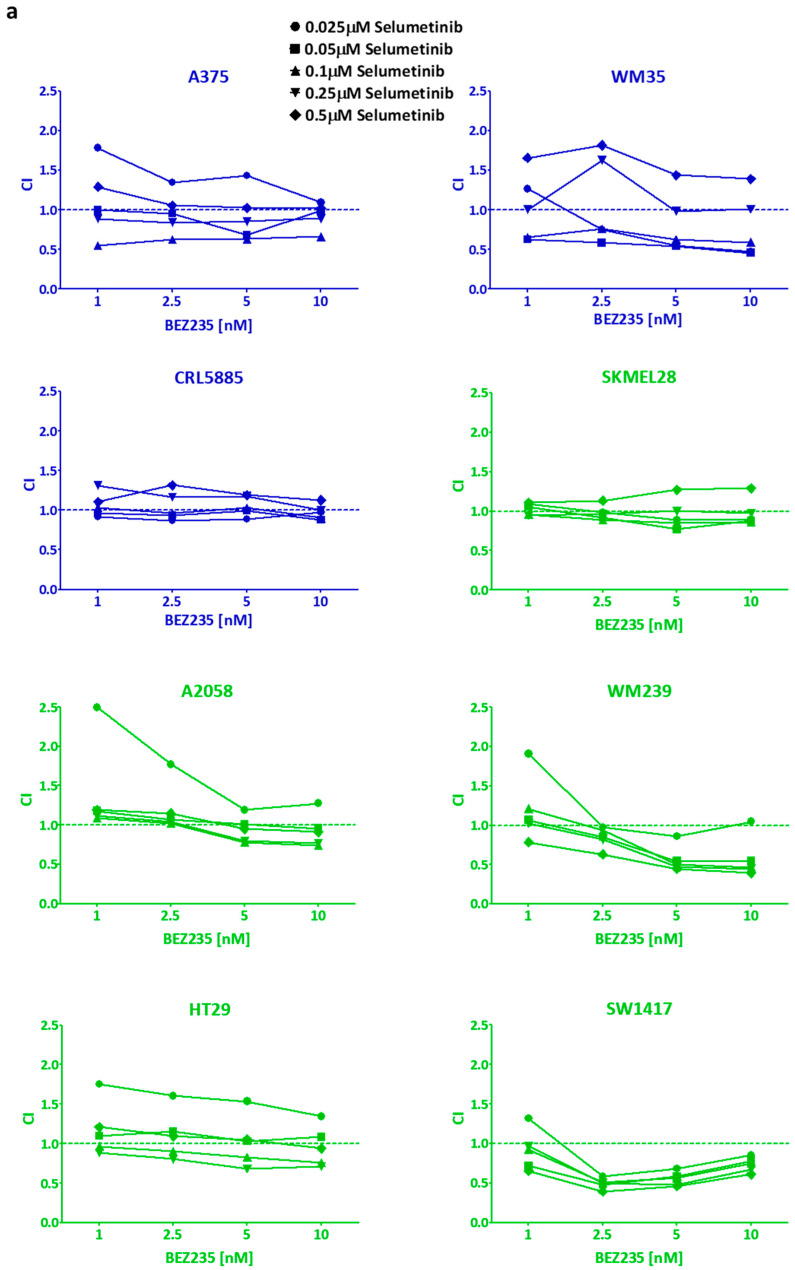
Combinatory effect of selumetinib + BEZ235 on the cell lines for 10 days. (**a**) Combination index (CI) was calculated from long term (10 days) clonogenic assay results upon treatment with a combination of selumetinib and BEZ235 in different concentrations. CI < 1, CI ≈ 1, and CI > 1 mean synergistic, additive, and antagonistic effect, respectively. In most of the cell lines, the combination treatments were closely additive except for WM239 and SW1417, where synergy was detected. Data is from at least three independent experiments. (**b**) The average CI values of the cells. Data is shown as mean ± SEM from at least three independent experiments. Combination index in WM239 and SW1417 cell lines was significantly lower than 1. Asterisks mean a significant difference between CI ≈ 1 and the given CI value of the cell by * *p* < 0.05.

**Figure 4 ijms-21-07649-f004:**
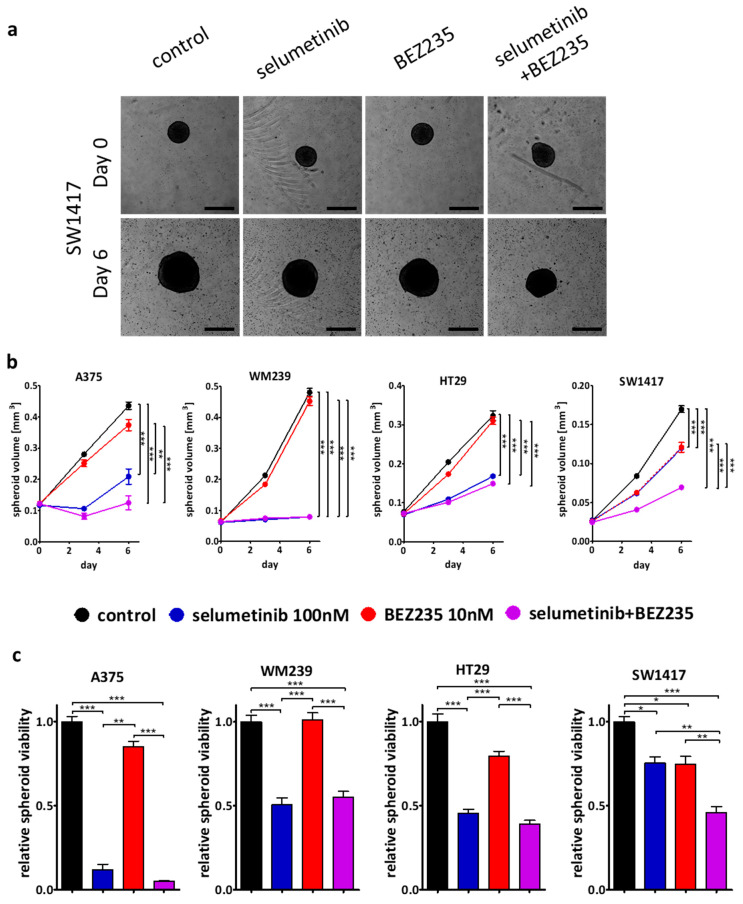
3D spheroid growth inhibition results upon treatment with single compounds or combination for 6 days. A375, WM239, HT29, and SW1417 cell lines were involved into 3D investigations. (**a**) Representative images of SW1417 spheroids from treatment day 0 and day 6. Scale bar indicate 500 µm. (**b**) Spheroid volume was detected by taking pictures during the treatment and (**c**) spheroid proliferation by CCK8 at the end of the experiment. Colors indicate the therapy as blue, red means selumetinib, BEZ235 and purple indicates the combination of them, respectively. The combination treatment was significantly more effective than single treatments only in the case of SW1417 spheroids. Data is shown as mean ± SEM from three independent experiments. Asterisks mean a significant difference between the treatment groups by * *p* < 0.05, ** *p* < 0.01 and *** *p* < 0.001.

**Figure 5 ijms-21-07649-f005:**
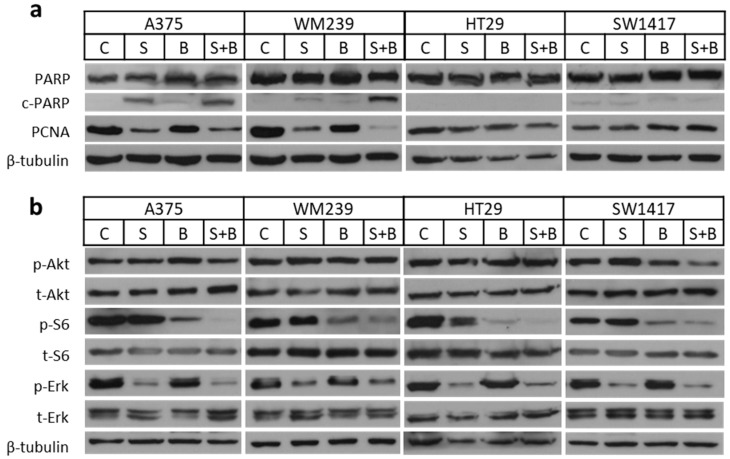
Protein expression investigation by Western blot upon treatment with selumetinib and/or BEZ235. (**a**) For c-/PARP and PCNA detection 100nM selumetinib (S) and/or 10nM BEZ235 (B) was applied for 48 hours. Apoptosis induction (c-PARP) was detected upon treatment with selumetinib and selumetinib + BEZ235 in case of A375 and WM239. Decreased expression of PCNA upon selumetinib or combination treatment was detected in A375 and WM239 cells. (**b**) For signaling pathway element detection (Akt, Erk, S6), 50nM of selumetinib, 5nM of BEZ235 or the combination of them was used for 4 hours. In all four cell lines, S6 and Erk activation decreased upon treatment with BEZ235/selumetinib + BEZ235 and selumetinib/selumetinib + BEZ235, respectively. However, Akt activation decrease was detected only in case of SW1417 cell line. Blots are representative pictures from three independent experiments.

**Figure 6 ijms-21-07649-f006:**
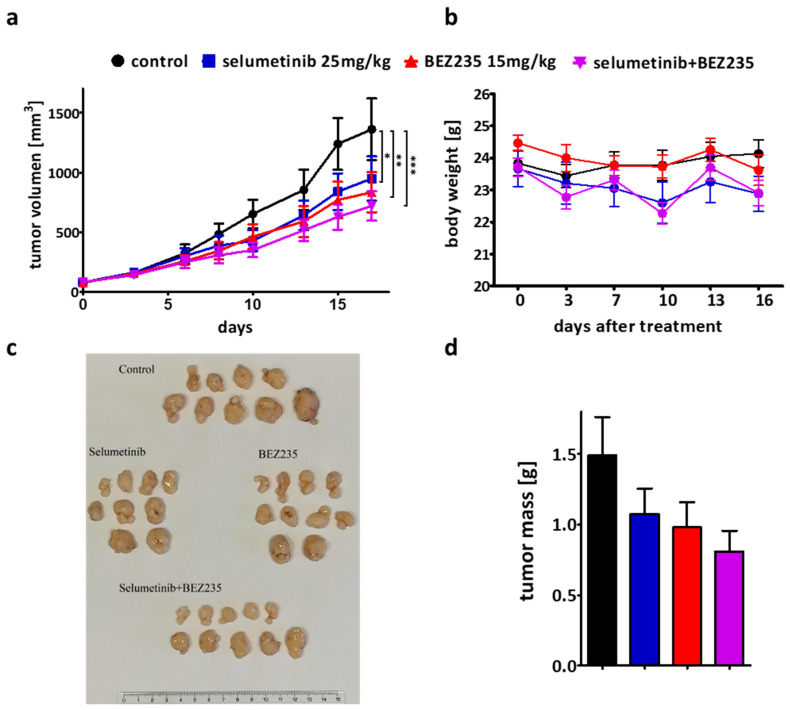
Subcutaneous xenograft model was established in NOD-SCID mice from SW1417 cells to determinate the effect of the single treatment and combinational therapy. Daily oral treatment of selumetinib (25 mg/kg) and BEZ235 (15 mg/kg) and the combination of them was used for 17 days. (**a**) Graph indicates tumor volume growth. A stronger effect of the combination treatment was detected compare to the single agents from the first measurement of the tumor volumes (third day). (**b**) Changes in the body weight during the therapy. (**c**) Pictures of the tumors from the animal experiment upon treatment with the indicated drug for 17 days. (**d**) Tumor mass upon 17-day treatment with vehicle (black), selumetinib (blue), BEZ235 (red), or combination (purple). The combination treatment had the highest inhibitory effect on tumor mass compare to the single treatments. Data is shown as average ± SEM from *n* = 9−10 groups. Asterisks indicate a significant difference between the treatment groups and the control by * *p* < 0.05, ** *p* < 0.01 and *** *p* < 0.001.

**Table 1 ijms-21-07649-t001:** Mutational status of the cell lines (based on CCLE, COSMIC, and ExPASy databases).

Cell Line	Tissue	BRAF	PI3K/PTEN	RAS
A375	melanoma	V600E	wild type	wild type
WM35	melanoma	V600E	wild type	wild type
CRL5885	lung	G466V	wild type	wild type
SKMEL28	melanoma	V600E	PTEN T167A	wild type
A2058	melanoma	V600E	PTEN null	wild type
WM239	melanoma	V600D	PTEN null	wild type
HT29	colon	V600E	PI3KCA P449T	wild type
SW1417	colon	V600E	delPI3KR1 [70]	wild type

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
