# Peer review of "Horizontal Combination of MEK and PI3K/mTOR Inhibition in BRAF Mutant Tumor Cells with or without Concomitant PI3K Pathway Mutations"

_ijms, 2020, doi:10.3390/ijms21207649_

Round 1

Reviewer 1 Report

This is an interesting study to investigate the targeted therapy in various BRAF mutated cell lines with or without PI3K/Akt alteration.

However, some major drawbacks should be addressed to make me reject this basic study.

   For the cell-line part:

  1. CRL5885 harboring G466V mutation which belongs to class III mutation. This kind of mutation is completely different from class I BRAF mutation (like V600E). The authors should notice this point and cannot consider they are the same.

  1. Colon cancer with BRAF mutation is much different from other cancers with BRAF mutation due to overexpressed EGFR. Inhibiting MAPK would enhance the paradoxical activation of EGFR so this could explain why two cell lines (HT29 and SW1417) are more resistant to MEKi. In addition, a study from lung cancer has shown the patients with BRAF mutation coexisting with PI3K alterations had a poor response to targeted therapy (dabrafenib and trametinib). So various cancer with BRAF V600E mutation really did have different biological meanings from the clinical observation.

  1. It is not necessary and reasonable to merge cell lines with similar genetic alterations together for comparison as they are not isogenic cell lines. (Figure 1C,1D, 2C, 2D). The authors should generate isogenic cell lines to confirm their hypothesis.

For BEZ 235, from the clinical point of view, it is totally abandoned for clinical use due to unfavorable safety profile and unpredictable bioavailability led to the sponsor’s decision to halt the development of BEZ235 in all oncology indications. Why does the investigator to choose it for the experiment.

Minor

  1. Why did the authors use a different concentration of selumetinib and BEZ235 for Figures 5A and 5B?
  2. The dose and time point (48h) used in figure 5A were usually. If the authors used so high concentration for combination, very few cells can be left after 48 hours of treatment for western blotting according to other results. Flow should be done to evaluate apoptosis.  

Reviewer 2 Report

Here the authors investigated the MEK inhibitor selumetinib and PI3K/mTOR dual inhibitor BEZ235 alone and in combination in BRAF-only mutant and BRAF+PI3K/PTEN double mutant cancer cells using short- and long- term 2D viability assays, spheroid assays and immunoblots. Their study demonstrates that the concomitant mutations in RAS/RAF and PI3K/Akt  pathway does not unequivocally predict synergistic effect of the horizontal combination of inhibitors of MEK and PI3K/mTOR. Overall the this is an interesting study, however, I do suggest the authors add a section on combination therapy and the success of this approach for breast cancer therapy in the Discussion/Introduction. Useful references for this include

1. Reza Bayat Mokhtari, Tina S. Homayouni, Narges Baluch, Evgeniya Morgatskaya, Sushil Kumar, Bikul Das and Herman Yeger (2017). Combination therapy in combating cancer. Oncotarget. 10.18632/oncotarget.16723

2. Dhananjay Huilgol, Prabhadevi Venkataramani, Saikat Nandi, Sonali Bhattacharjee (2019). Targeting transcription factors that govern development and disease: An achilles heel for cancer therapeutics. Genes. 10.3390/genes10100794

3. Sonali Bhattacharjee and Saikat Nandi (2018) Rare Genetic Diseases with Defects in DNA Repair: Opportunities and Challenges in Orphan Drug Development for Targeted Cancer Therapy. Cancers. 10.3390/cancers10090298

Round 2

Reviewer 1 Report

After reviewing the revised version of the paper, I am satisfied with the response fro the investigators.

Reviewer 2 Report

The authors have responded to my comments sufficiently.